# Use of the Airstretcher with dragging may reduce rescuers' physical burden when transporting patients down stairs

Yutaka Takei[1]*, Eiji Sakaguchi[2], Koichi Sasaki[2], Yoko Tomoyasu[2], Kouji Yamamoto[2], Yasuharu Yasuda[2]

**1** Department of Emergency Medical Sciences, Niigata University of Health and Welfare, Niigata, Japan, **2** Department of Prehospital Emergency Medical Sciences, Hiroshima International University, Hiroshima, Japan

* yutaka-takei@nuhw.ac.jp

**Data Availability Statement:** All relevant data are within the manuscript and its Supporting Information files.

## Abstract

Transporting patients down stairs by carrying is associated with a particularly high fall risk for patients and the occurrence of back pain among emergency medical technicians. The present study aimed to verify the effectiveness of the Airstretcher device, which was developed to reduce rescuers' physical burden when transporting patients by dragging along the floor and down stairs. Forty-one paramedical students used three devices to transport a 65-kg manikin down stairs from the 3rd to the 1st floor. To verify the physical burden while carrying the stretchers, ratings of perceived exertion were measured using the Borg CR10 scale immediately after the task. Mean Borg CR10 scores (standard deviation) were 3.6 (1.7), 4.1 (1.8), 5.6 (2.4), and 4.2 (1.8) for the Airstretcher with dragging, Airstretcher with lifting, backboard with lifting, and tarpaulin with lifting conditions, respectively ($p < 0.01$). Multiple comparisons revealed that the Airstretcher with dragging condition was associated with significantly lower Borg CR10 scores compared with the backboard with lifting condition ($p < 0.01$). When the analysis was divided by handling position, estimated Borg CR10 values (standard error) for head position were 4.4 (1.3), 2.9 (0.9), 3.2 (0.8), and 4.0 (1.1) for the Airstretcher with dragging, Airstretcher with lifting, backboard with lifting, and tarpaulin with lifting conditions, respectively, after adjusting for participant and duration time (F = 1.4, $p < 0.25$). The estimated Borg CR10 value (standard error) for toe position in the Airstretcher with dragging condition was 2.0 (0.8), and the scores for the side position were 4.9 (0.4), 6.1 (0.3), and 4.7 (0.4) for the Airstretcher with lifting, backboard with lifting, and tarpaulin with lifting conditions, respectively, after adjusting for participant and duration time (F = 3.6, $p = 0.02$). Transferring a patient down stairs inside a house by dragging using the Airstretcher may reduce the physical burden for rescuers.

## Introduction

Emergency medical technicians (EMTs) respond to emergency situations immediately, treat patients in a timely fashion, make decisions about which hospitals to transport patients to, and

**Funding:** The authors received no specific funding for this work.

**Competing interests:** The authors have declared that no competing interests exist.

transfer patients quickly. Some studies reported the frequent occurrence of adverse events during their activities, including back pain among EMTs and fall events for patients [1–4]. An observational study reported that the occurrence of back pain during work was linked to EMTs leaving their jobs or being transferred to a different role [5].

Although powered stretchers with automated loading systems may be effective for preventing musculoskeletal disorders among EMTs and fall events for patients when loading patients into ambulances, the use of manual stretchers is still the most common method of patient transport in Japan [6, 7]. Furthermore, soft stretchers must be used in some situations because corridors and stairs inside houses in Japan are particularly narrow compared with those of other countries.

A variety of transferring instruments are available, including the tarpaulin and rescue sheet (Ferno Japan, Inc., Japan), which are commonly used inside houses. The rigid backboard is a motion restriction device designed for cases of spinal injury and is commonly used in callouts when transferring from the scene to the main stretcher. However, during carrying, transporting patients down stairs is associated with a particularly high fall risk for patients and the occurrence of back pain among EMTs. In a previous study, we identified the occurrence of falls and back pain during on-scene activities as being linked to severe accidents in some cases [3].

The Airstretcher® (Airstretcher, Inc., CA, USA) was recently developed to transport patients by dragging along the floor and stairs. This device has the benefit of reducing the risk of fall events by avoiding the need to lift the patient. We hypothesized that the Airstretcher may decrease the physical burden of EMTs during callouts compared with other methods. The purpose of the present study was to assess the effectiveness of this device for decreasing rescuers' physical burden when carrying patients down stairs.

## Materials and methods

This study was approved by the Niigata University of Health and Welfare Ethics Committee (18452–200717). The study's objective, significance, methods, and the process of opting out were explained to participants in writing and verbally beforehand. The participants provided written informed consent.

### Participants

The participants were 41 third-year paramedical students. Participants had acquired basic paramedical knowledge and had experience with the use of soft stretchers including a tarpaulin and backboards in simulation training.

Before the study, participants attended a 45-minute lecture about how to use soft stretchers, and a 45-minute training session involving the transport of a 65-kg manikin down stairs using an Airstretcher with lifting and with dragging.

### Instruments (Fig 1)

We compared the use of three types of instruments for transporting patients down stairs: 1) the Airstretcher, 2) the backboard (Ferno Japan, Inc., Japan) and 3) the tarpaulin stretcher. Each stretcher was loaded with a 65-kg manikin, and groups of three participants were instructed to carry it from the 3rd floor to the 1st floor. Because the students had already been grouped into classes at University, we decided to recruit each group to participate in the study. Participants used the stretchers in the order in which they were accustomed to using them. Therefore, each group carried the manikin using the tarpaulin first, followed by the backboard, the Airstretcher by lifting, and the Airstretcher by dragging. When carrying the manikin, participants selected their own holding position between the head/side/toe sides. When transporting the manikin using the Airstretcher carrying by dragging, two participants carried the

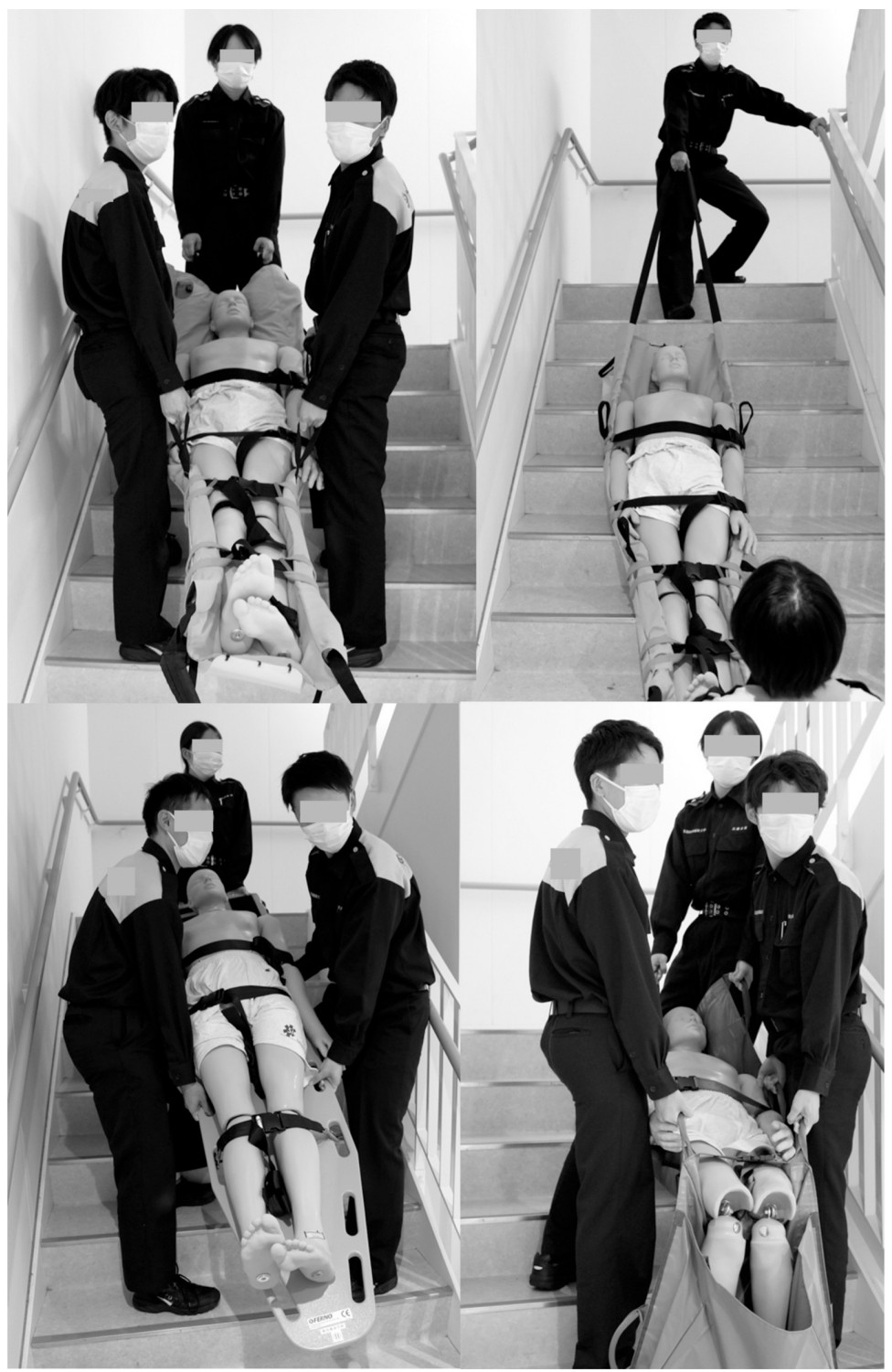

**Fig 1. Instruments.** The upper left image shows the use of the Airstretcher by lifting. The upper right image shows the use of the Airstretcher by dragging without lifting. The lower left image shows the backboard being carried by lifting. The lower right image shows the tarpaulin being carried by lifting.

instrument, using wide shoulder slings on the head and toe sides of the stretcher. Eleven participants did not carry the manikin when they used the Airstretcher by dragging, because the device does not require more than two operators. Participants decided themselves in each group who would not operate the Airstretcher by dragging. For this reason, we excluded the data of 11 participants from the analysis. Participants were given a sufficient rest period between each carrying session.

Finally, the comparative performance of each instrument was examined using a questionnaire survey.

**The Airstretcher (carrying by lifting).**　The Airstretcher (www.airstretcher.jp/) has a Cappy original mat produced by Vinal Technology, Inc., and has a length of 192 cm × width of 63 cm × height of 4.5 cm, and a weight of 5.5 kg. The bottom is made of special thermoplastic polyolefin or polyethylene plastic, with a thickness of 2 mm. The head and toe sides of the stretcher have wide slings which can be worn over the shoulder. The instrument has four fixing belts and six gripping points, including the head and toe sides. EMTs generally carry patients by lifting with three or more personnel. The Airstretcher is a registered trademark of Cappy International, Inc., Japan.

**The Airstretcher (carrying by dragging, without lifting).**　The Airstretcher can be used to transport a patient by dragging by a single operator. In Japan, this stretcher has already used to hospitals, nursing homes, ambulance services, schools, police departments, the Self-Defense Forces, and other institutions. By dragging the Airstretcher, it is relatively easy to transport a patient down narrow stairs with one or two operators. This instrument can be used on any surface, including asphalt, gravel, or iron floor plates. When the EMT opens the air valve on the stretcher, air automatically flows into the mattress.

**The backboard (carrying by lifting).**　The high-tech backboard (http://www.ferno-jp.com/) is made of acrylonitrile butadiene styrene plastic, which prevents blood and body fluids from absorbing into the interior and allows easy cleaning. The backboard is commonly used for restricting spinal movement of injured persons, mainly in trauma cases. This instrument has length of 183 cm × width of 41 cm × height of 4.5 cm, a weight of 5.9 kg, and a maximum load of 159 kg.

**The tarpaulin (carrying by lifting).**　This instrument enables transport in narrow stairways, arund tight bends, and in elevators. It is equipped with an anchoring belt to help the patient remain calm. The instrument is constructed from a waterproof tarpaulin fabric and has a length of 180 cm × width of 48 cm × height of 1 cm, and a weight of 1.8 kg. The instrument has two fixing belts and four gripping points, including the head and toe sides.

## Evaluation

To verify the physical burden of carrying the stretchers, ratings of perceived exertion were measured using the Borg CR10, a category ratio (CR) scale (Table 1) from 1–10, that was completed immediately after each task. Additionally, pulse rates after the task in all participants were measured [8] using fingertip pulse oximeter. The Borg CR10 scale, pulse rates and the carrying duration were recorded and put it into the database by researchers.

## Data analysis

In the univariate analyses of continuous variables, one-way analysis of variance was applied. Multiple comparisons were calculated using the Tukey-Kramer test. To examine the comparative performance among the instruments, we used multiple least squares regression analysis. With a significance level of $p = 0.05$ and a power value of 0.8 at an effect size of 0.1, we estimated that the study needs 27 participants in each group. All data were analyzed using JMP

**Table 1. The Borg CR10 scale [8].**

| Score | Level of exertion |
|:---:|:---|
| 0 | No exertion at all |
| 0.5 | Very, very slight (just noticeable) |
| 1 | Very slight |
| 2 | Slight |
| 3 | Moderate |
| 4 | Somewhat severe |
| 5 | Severe |
| 6 | |
| 7 | Very severe |
| 8 | |
| 9 | Very, very severe (almost maximal) |
| 10 | Maximal |

CR: category ratio

software (version 14.3; SAS Institute, Cary, NC, USA). For each analysis, the null hypothesis was evaluated with a two-sided significance level of $p < 0.05$.

# Results

Thirty-six male and five female students participated in the study. Their mean height (standard deviation [SD]) was 169.8 (6.2) cm, and their mean weight (SD) was 62.9 (8.5) kg.

## Borg CR10 scale (Fig 2)

**Overall comparisons.** Mean (SD) scores on the Borg CR10 scale were 3.6 (1.7), 4.1 (1.8), 5.6 (2.4) and 4.2 (1.8) for the Airstretcher with dragging [AD], Airstretcher with lifting [AL], backboard with lifting [BL] and tarpaulin with lifting [TL] conditions, respectively. The scores were highest when participants carried a manikin using the TL and lowest when they used the AD ($p < 0.01$). Multiple comparisons revealed that the AD condition was associated with significantly lower Borg CR10 scores compared with the BL condition ($p < 0.01$). The TL condition was associated with significantly lower Borg CR10 scores compared with the BL condition ($p < 0.01$).

**Sub-group analysis.** Because the multiple least square's regression analysis revealed that instrument and handling position exhibited a significant interaction ($p < 0.01$, Table 2), we applied sub-group analysis.

*Interaction between the Borg CR10 scale and other factors.* As shown in Fig 3A, estimated values of the Borg CR10 scale (standard error [SE]) were 2.8 (0.6), 4.0 (0.4), 4.9 (0.4) and 4.5 (0.4) for the AD, AL, BL and TL conditions, respectively, after being adjusted for participant, handling position and duration time (F = 3.1, $p < 0.01$). Multiple comparisons revealed a significant difference between the AD and the BL conditions ($p = 0.02$).

*Head position.* When the analysis was divided by handling position (Fig 3B), estimated values (SE) of the Borg CR10 scale of head position were 4.4 (1.3), 2.9 (0.9), 3.2 (0.8) and 4.0 (1.1) for the AD, AL, BL and TL conditions, respectively, after adjusting for participant and duration time (F = 1.4, $p < 0.25$). Multiple comparisons revealed no significant differences among instruments.

*Toe and side positions.* As shown in Fig 3C, the estimated value on the Borg CR10 scale (SE) of toe position in the AD condition was 2.0 (0.8), and the scores for the side position were 4.9

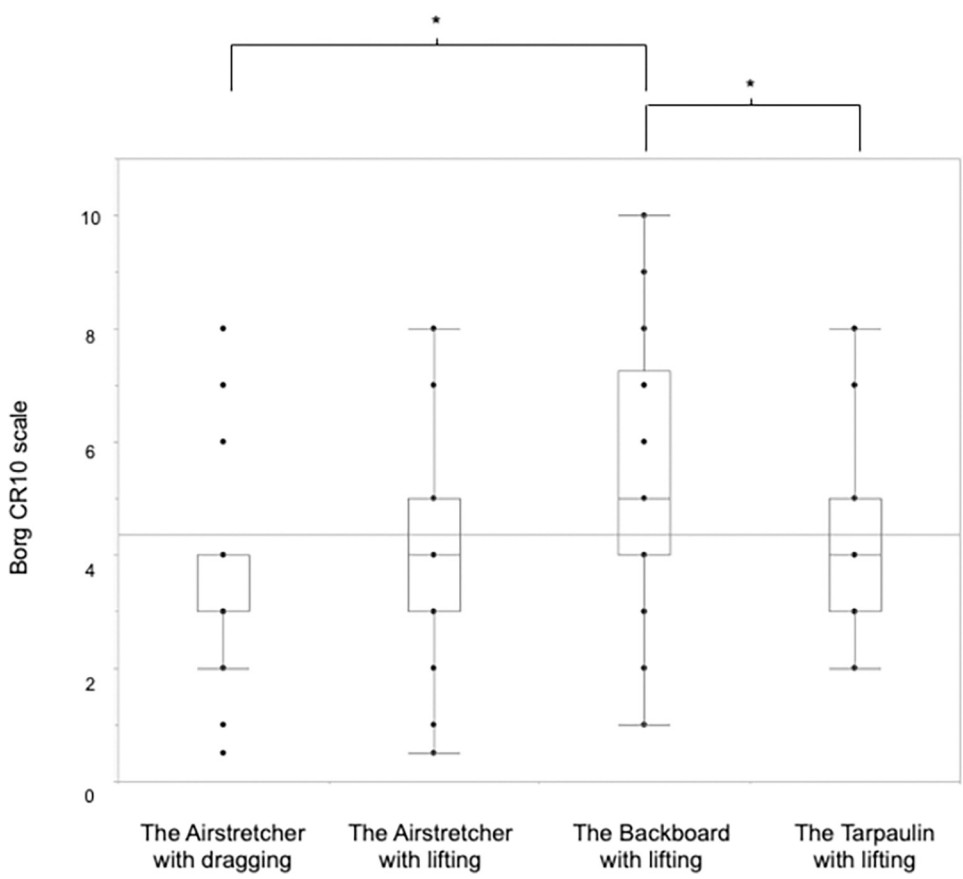

**Fig 2. The Borg CR10 scale.** * $p < 0.01$, Tukey-Kramer's HSD test.

(0.4), 6.1 (0.3) and 4.7 (0.4) for the AL, BL and TL conditions, respectively, after adjusting for participant and duration time (F = 3.6, $p = 0.02$). Borg CR10 scores were lower in the AD condition compared with those in the AL condition ($p = 0.02$) and the BL condition ($p < 0.01$). Scores were lower in the TL condition compared with those in the BL condition ($p = 0.04$).

## Pulse rates (Fig 4)

Pulse rates after the task were 136.4 bpm, 132.5 bpm, 135.5 bpm and 127.9 bpm for the AD, AL, BL and TL conditions, respectively. However, the univariate analyses did not reveal any significant differences in pulse rates among the four carrying methods ($p = 0.36$).

**Table 2. Multiple least squares regression analysis.**

|  | *p*-value |
|---|---|
| Interaction (instrument and position) | < 0.01 |
| Instruments | < 0.01 |
| Position (head or side/toe) | < 0.01 |
| Participants | 0.02 |
| Duration | 0.04 |

Summary of fit: root mean squared error (RMSE) = 1.65, $R^2$ = 0.59.

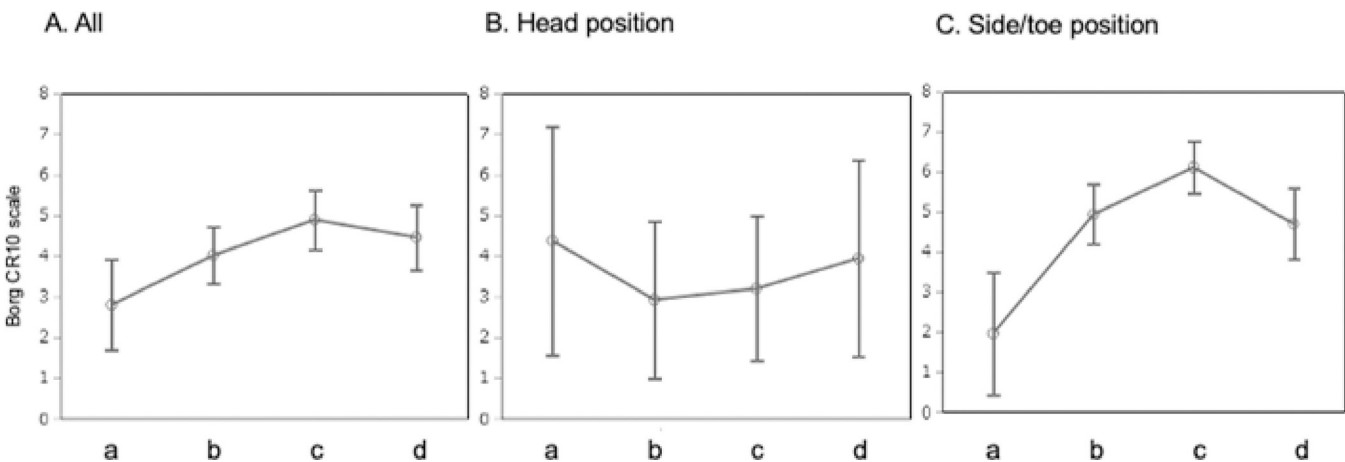

**Fig 3. The Borg CR10 scale after adjustment for other factors.** a: The Airstretcher with dragging, b: The Airstretcher with lifting, c: The Backboard with lifting, d: The Tarpaulin with lifting.

## Carrying duration (Fig 5)

Mean (SD) carrying durations were 83.7 s (17.1), 51.1 s (6.2), 54.9 s (9.2) and 45.3 s (5.1) in the AD, the AL, the BL and the TL conditions, respectively ($p < 0.01$). The duration in the AD condition was significantly longer compared with the other conditions ($p < 0.01$ for each). The duration in the TL condition was significantly shorter than that in the BL condition ($p < 0.01$).

**Questionnaire surveys.** Finally, we asked the participants which instruments were the most and least physically demanding to use. Thirty-two (78.0%) reported that the BL was the

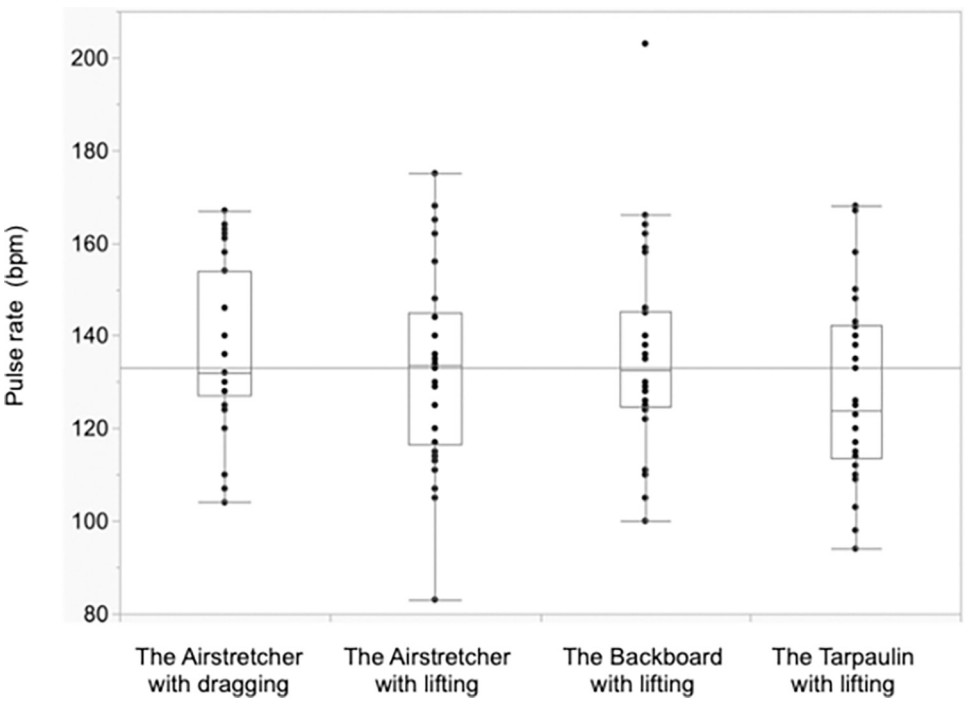

**Fig 4. Pulse rate after carrying.**

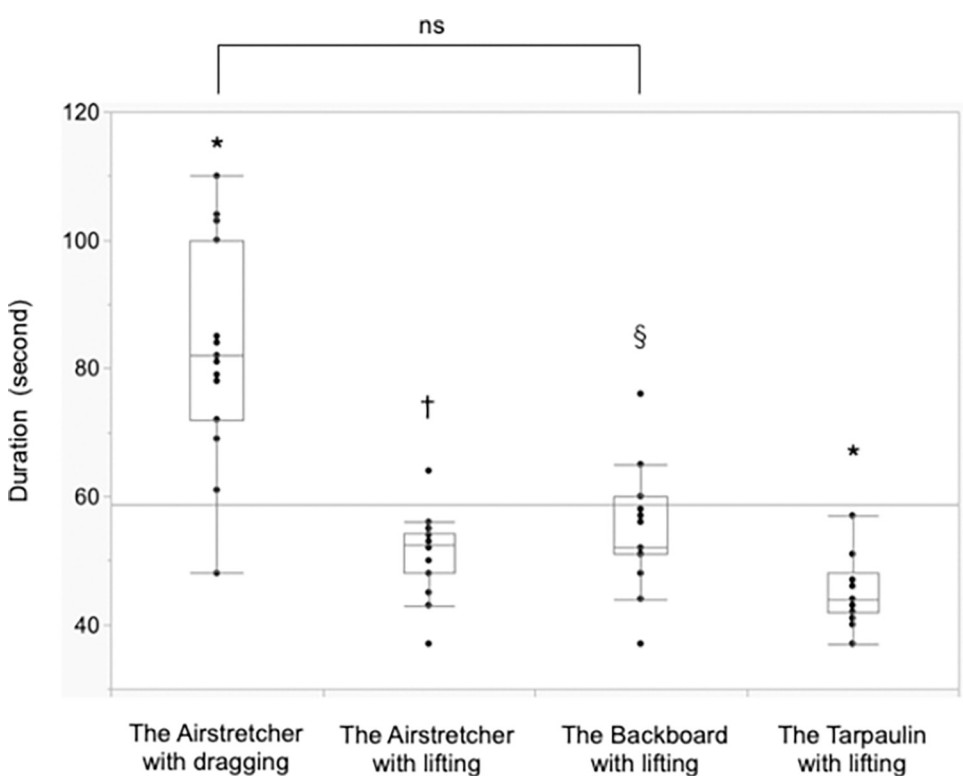

**Fig 5. Duration of carrying.** * Compared with others ($p < 0.01$). † Compared with the Airstretcher with dragging ($p < 0.01$) and the Tarpaulin with lifting ($p < 0.05$). §Compared with the Airstretcher with dragging ($p < 0.01$) and the Tarpaulin with lifting ($p < 0.01$).

most physically demanding, and seven (17.1%) reported that the TL was the most physically demanding. In addition, 21 (51.2%) participants reported that the AD was the least physically demanding, and 17 (41.5%) participants reported that the TL was the least physically demanding.

## Discussion

The present study demonstrated the superiority of using the Airstretcher by dragging for when rescuers transport a patient from an upper floor to a lower floor via the stairs in terms of physical burden, compared with other methods. More than half of the participants reported that using the Airstretcher by dragging involved the lowest physical burden, even though this was their first experience using the Airstretcher instrument. In addition, ratings of perceived exertion were lowest for the Airstretcher with dragging condition, compared with the other methods. Thus, the Airstretcher may have benefits over other soft stretchers for decreasing physical burden among EMTs, despite the tarpaulin and the backboard being most commonly used for carrying patients inside a house.

The optimal method for carrying patients in pre-hospital settings has not been clarified in previous studies. However, one study reported that lifting techniques with soft stretchers have failed to reduce lifting-related injuries and are unsafe for providers and patients [9]. Our previous study confirmed that fall events among patients often occur during carrying on the scene, by analyzing the database of Fire and Disaster Management Agency in Japan. In Japan, soft stretchers are commonly used, and may be linked to falls among patients. In the current study, we explored the performance of a newly developed patient-carrying instrument designed to

decrease the fall risk and physical burden. The current findings suggest that using the Airstretcher with dragging may decrease the physical burden for EMTs.

As an index of physical burden, we compared participants' ratings of perceived exertion and pulse rate after the completion of each task. The carrying duration when using the Airstretcher with dragging was longer than that in the other conditions. However, estimated values of the Borg CR10 score tended to be lower in the Airstretcher with dragging condition compared with the other conditions. Participants were more familiar with handling the backboard and the tarpaulin compared with the Airstretcher and may have known the optimal amount of hand pressure to use while handling both instruments, but not the Airstretcher. The bottom of the Airstretcher is made of a plastic material, which is a smooth surface that slides easily. Duration of carrying may have increased as they tried to grasp the stretcher firmly to prevent it from sliding down the stairs. Attempting to operate the Airstretcher carefully may have been linked with the longer carrying duration we observed. It should be noted that Borg CR10 scores exhibited a wide range in the Airstretcher with dragging condition. An experimental study revealed that increased muscle oxygenation for experienced subjects is less than that for novice subjects [10–13]. Therefore, muscle oxygenation may differ between trained and untrained individuals. Further training in the handling of the Airstretcher may enable safer and less stressful patient transport.

## Limitations

The current study involved several limitations. First, this study sought to identify the most effective instrument for transporting patients down stairs in a pre-hospital setting. However, the operators were students rather than EMTs with work experience. The Airstretcher, which is a newly designed instrument, is not widely used in emergency medical services in Japan. Although we assume that EMTs are likely to have similar preferences to paramedical students, this assumption remains to be verified.

Second, we did not compare the Airstretcher with the Stairchair, which is commonly used internationally. However, use of the Stairchair involves a risk of damaging surfaces inside a house, and stairs inside Japanese houses are typically narrow and constructed from wood. Furthermore, a flat position is typically more suitable in situations where a patient is unconscious. For these reasons, we did not examine this device in the current study.

Finally, the characteristics of the operator, including sex, weight and height may affect the outcome. These effects have been reported in previous studies [14, 15]. The sample size in the current study was larger than that in several other studies of this issue. Therefore, we believe that the present results are meaningful.

## Conclusion

Transferring a patient down stairs inside a house by dragging using the Airstretcher may reduce the physical burden for rescuers. Further verification by EMTs is necessary.

## Acknowledgments

We would like to thank all the participants for their great cooperation.

## Author Contributions

**Conceptualization:** Yutaka Takei.

**Data curation:** Yutaka Takei, Eiji Sakaguchi.

**Formal analysis:** Yutaka Takei.

**Investigation:** Yutaka Takei, Eiji Sakaguchi.

**Methodology:** Yutaka Takei.

**Project administration:** Yutaka Takei.

**Resources:** Yutaka Takei.

**Supervision:** Yutaka Takei, Koichi Sasaki, Yoko Tomoyasu, Kouji Yamamoto, Yasuharu Yasuda.

**Writing – original draft:** Yutaka Takei, Eiji Sakaguchi.

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
