## [Decision Letter · Decision Letter 0]

10 Jul 2022

PONE-D-21-35282Use of the Airstretcher with dragging may reduce rescuers’ physical burden when transporting patients down stairsPLOS ONE

Dear Dr. Takei,

Thank you for submitting your manuscript to PLOS ONE. After careful consideration, we feel that it has merit but does not fully meet PLOS ONE’s publication criteria as it currently stands. Therefore, we invite you to submit a revised version of the manuscript that addresses the points raised during the review process.

The reviewers concede that the Airstretcher may reduce the physical burden on EMT personnel compared to more traditional measures, although the current study design does present a few minor limitations. In a revision of the manuscript we would be keen on seeing the recommendations and comments f both reviewers tackled thoroughly.

We look forward to receiving your revised manuscript.

Kind regards,

Denis Alves Coelho, PhD

Academic Editor

PLOS ONE

Journal Requirements:

a) Did participants provide their written or verbal informed consent to participate in this study?

6. We note that Figure 1 includes an image of a participant in the study. 

Reviewers' comments:

Reviewer's Responses to Questions

**Comments to the Author**

1. Is the manuscript technically sound, and do the data support the conclusions?

Reviewer #1: Yes

Reviewer #2: Yes

2. Has the statistical analysis been performed appropriately and rigorously? 

Reviewer #1: No

Reviewer #2: Yes

3. Have the authors made all data underlying the findings in their manuscript fully available?

Reviewer #1: Yes

Reviewer #2: Yes

4. Is the manuscript presented in an intelligible fashion and written in standard English?

Reviewer #1: Yes

Reviewer #2: Yes

5. Review Comments to the Author

Reviewer #1: Comments

This manuscript reports the assessment of the effectiveness of the Airstretcher® device for decreasing emergency medical technicians’ physical burden when carrying patients downstairs. The Airstretcher was compared to two other instruments for transporting patients downstairs, namely the backboard and the tarpaulin stretcher. Please find below a few comments for your consideration. Line numbers in comments refer to the PDF file generated after manuscript submission.

Major comments

1. Lines 71-79. It is unclear whether all participants tested all 3 instruments (i.e. a repeated-measurement analysis), or if they were assigned to groups (i.e., independent-group analysis). I assume the latter was performed, but even in this case there is no description on how group allocation was performed (e.g. randomization?).

2. Lines 82-84. What was the order of testing the three instruments, fixed or random? What was the rationale for the choice of sequence?

3. Lines 88-89. It is unclear what is meant by ‘Most of the participants carried the manikin three times using the three types of stretcher’. How were these 11 participants selected? How the three repetitions for each instrument were aggregated (e.g. mean, smallest effort)?

4. Lines 130-132. Given the adopted scale (modified Borg scale) and vital sign (hear rate), I suggest using ‘perceived exertion’ rather than ‘physical burden’ across the manuscript.

5. Lines 133-134. How did you record the duration of the task?

6. Lines 136-137. Following comment #1, it is unclear what ANOVA was applied?

Minor comments

1. Abstract. Consider reporting SD alongside the means or the overall effect of the omnibus ad hoc tests (F-test and eta-sq values) rather than mean values with p values alone.

2. Lines 108-110. Why is it important to report the number of sold units in Methods section?

Reviewer #2: The submitted paper from Takei et al. proposes to investigate the potential benefits of using a Airstretcher – a newer device to aid EMT personnel when transporting incapacitated individuals – compared to traditional devices. In the experiment itself, students in EMT training performed the task of transporting a mannikin using either the Airstretcher or one of the more traditional devices. The outcomes measures included Borg’s RPE reporting, heart rate, and questionnaire responses. As reported by the authors, the Airstretcher may reduce the physical burden on EMT personnel compared to more traditional measures, although the current study design does present a few minor limitations.

1. The introduction was well-written and focused on the key aspects of the relevant background information for the study. One point of minor confusion may be lines 56-58, where the authors discuss the potential for falls, and how dropping the patient may lead to severe injuries. I would suggest including a sentence or clarifying here – given that incidence of drops/falls is not part of the present study, how does this relate to the physical burden of traditional devices versus the Airstretcher? Would the Airstretcher likely have a lower drop rate?

2. In the Methods Section (lines 77-79), this portion appears to be a bit repetitive. The specific details provided in this paragraph could likely be combined with the above sentences, to reduce unnecessary repetition.

3. In the Methods Section (line 133), it may be helpful to clarify further how the pulse rate was measured. This may help address questions about variability in the pulse rate measurements, etc.

4. The Results Section was very thorough, and provided detailed information about the outcomes of the study. One minor concern throughout was the readability – while minor, it may help with the ‘flow’ of the results section to simplify the naming of the four conditions with acronyms. In lines 152-153 where the four conditions are listed, parentheses could be included after each condition with an abbreviation or acronym. This would help reduce some of the longer sentences/phrases later in the results section, and clarify the overall section for the reader.

5. Also in the Results Section, it may help to ‘break up’ the Borg RPE section with a few subheadings. This would clarify to the reader what specific aspect of the Borg RPE reporting that the current section is addressing.

6. The Discussion Section was very well done, and easy to follow. One minor point would be the sentence in lines 247-248 dealing with muscle oxygenation. For those familiar with the relationship between muscle oxygenation and physical exertion/workload, this sentence does provide further evidence of the physical demands of transporting patients. However, this sentence may be more clear if the authors expanded upon the interpretation of the findings that muscle oxygenation differs for trained versus untrained individuals, given that the nature of this paper may interest readers who are not as familiar with concepts such as skeletal muscle oxygenation.

7. The Limitations portion of the Discussion Section was well-written, and sets up future directions in a concise but meaningful way. This section was particularly well-handled.

8. One concern in the Figures – Figure 2 presents a box-and-whisker plot for the Borg RPE values. However, this Figure is a bit unclear given the alignment of the dots and variability of their placement. It may help clarify the figure to align the dots over the condition they represent, or choose some other way to represent the individual data.

6. PLOS authors have the option to publish the peer review history of their article (what does this mean?). If published, this will include your full peer review and any attached files.

Reviewer #1: **Yes: **Arthur de Sá Ferreira

Reviewer #2: No

---

## [Author Response · Author response to Decision Letter 0]

31 Jul 2022

Response to Reviewers

Dear Editors and Reviewers

Thank you very much for reviewing our manuscript and offering valuable advice.

We have addressed your comments with point-by-point responses and revised the manuscript accordingly.

We wish to express our appreciation to the Reviewers for their insightful comments, which have helped us significantly improve the paper.

Journal requirements

#0-1. Journal requirements

Reply to journal requirements

1) Our manuscript style has been changed according to PLOS ONE’s style requirements.

#0-2. Journal requirements

a) Did participants provide their written or verbal informed consent to participate in this study?

Reply to journal requirements

2) We added ethics statement in the first paragraph of Materials and methods as below:

- “The study’s objective, significance, methods, and the process of opting out were explained to participants in writing and verbally beforehand. The participants provided written informed consent.”

#0-3. Journal requirements

3. In your Data Availability statement, you have not specified where the minimal data set underlying the results described in your manuscript can be found. PLOS defines a study's minimal data set as the underlying data used to reach the conclusions drawn in the manuscript and any additional data required to replicate the reported study findings in their entirety… Upon re-submitting your revised manuscript, please upload your study’s minimal underlying data set as either Supporting Information files or to a stable, public repository and include the relevant URLs, DOIs, or accession numbers within your revised cover letter.

Reply to journal requirements

3) The data set has been uploaded as the Supporting Information file, and we mentioned it in the cover letter. Please confirm the file.

#0-4. Journal requirements

Reply to journal requirements

4) We would like to upload our data set as “Data Availability File” when we resubmit, not to provide repository information. Therefore, we wrote it in the covering letter.

#0-5. Journal requirements

Reply to journal requirements

5) We confirmed the ethics statement only appeared int the Methods section in the manuscript.

#0-6. Journal requirements

6. We note that Figure 1 includes an image of a participant in the study…

Reply to journal requirements

6) Thank you for pointing it out. We have masked individual faces in the photograph of Fig 1 so that they can not be identified. Please confirm Fig 1.

 

Reviewer 1 comments

#1-1. Reviewer-1

1. Lines 71-79. It is unclear whether all participants tested all 3 instruments (i.e. a repeated-measurement analysis), or if they were assigned to groups (i.e., independent-group analysis). I assume the latter was performed, but even in this case there is no description on how group allocation was performed (e.g. randomization?).

Reply to Reviewer

1) We wish to express our appreciation to the Reviewer for their insightful comments, which have helped us significantly improve the paper. We have revised the Methods section to establish a clearer method as follow:

- “Because the students had already been grouped into classes at university, we decided to recruit each group to participate in the study. Participants used the stretchers in the order in which they were accustomed to using them. Therefore, each group carried the manikin using the tarpaulin first, followed by the backboard, the Airstretcher by lifting, and the Airstretcher by dragging.”

#1-2. Reviewer-1

2. Lines 82-84. What was the order of testing the three instruments, fixed or random? What was the rationale for the choice of sequence?

Reply to Reviewer

2) Thank you for providing these insights. As mentioned above, because the students had already been grouped into classes at university, we decided to recruit each group to participate in the study. Participants used the stretchers in the order in which they were accustomed to using them. Therefore, each group carried the manikin using the tarpaulin first, followed by the backboard, the Airstretcher by lifting, and the Airstretcher by dragging.”. In addition, we have mentioned in the same section as below: 

- “When carrying the manikin, participants selected their own holding position between the head/side/toe sides. When transporting the manikin using the Airstretcher carrying by dragging, two participants carried the instrument, using wide shoulder slings on the head and toe sides of the stretcher. Eleven participants did not carry the manikin when they used the Airstretcher by dragging, because the device does not require more than two operators.”

#1-3. Reviewer-1

3. Lines 88-89. It is unclear what is meant by ‘Most of the participants carried the manikin three times using the three types of stretcher’. How were these 11 participants selected? How the three repetitions for each instrument were aggregated (e.g. mean, smallest effort)?

Reply to Reviewer

3) Thank you for providing these insights. We deleted the sentence “Most of the participants carried the manikin three times using the three types of stretcher”, and added following sentences: 

- “Eleven participants did not carry the manikin when they used the Airstretcher by dragging, because the device does not require more than two operators. Participants decided themselves in each group who would not operate the Airstretcher by dragging.”

The question of "how the three repetitions for each instrument were aggregated" has been mentioned the sentences in the Evaluation part. However, we revised the sentences as, 

– "...that was completed immediately after each task. Additionally, pulse rates after the task in all participants were measured, and the Borg CR10 scale, pulse rates and the carrying duration were recorded by researchers."

#1-4. Reviewer-1

4. Lines 130-132. Given the adopted scale (modified Borg scale) and vital sign (hear rate), I suggest using ‘perceived exertion’ rather than ‘physical burden’ across the manuscript.

Reply to Reviewer

4) This is a valid assessment of your suggestion; however, in the fields of pre-hospital emergency care as we referenced several articles, same studies using same methods called/defined current evaluation as "physical burden". We would like to standardize a definition as "physical burden". However, if you disagree with this, we could change it from "physical burden" to "perceived exertion".

#1-5. Reviewer-1

5. Lines 133-134. How did you record the duration of the task?

Reply to Reviewer

5) As replied above, we revised/added the sentence “that was completed immediately after each task. Additionally, pulse rates after the task in all participants were measured [8], and the Borg CR10 scale, pulse rates and the carrying duration were recorded and put it into the database by researchers."

#1-6. Reviewer-1

6. Lines 136-137. Following comment #1, it is unclear what ANOVA was applied?

Reply to Reviewer

6) Thank you for providing these insights. We added a word in the sentence as below:

- “In the univariate analyses of continuous variables, one-way analysis of variance was applied.”

#1-7. Reviewer-1, as minor comments

1. Abstract. Consider reporting SD alongside the means or the overall effect of the omnibus ad hoc tests (F-test and eta-sq values) rather than mean values with p values alone.

Reply to Reviewer

7) Thank you for the comment. We have added reporting SDs and F values in the Abstract and Result section. Please confirm it.

#1-8. Reviewer-1, as minor comments

2. Lines 108-110. Why is it important to report the number of sold units in Methods section?

Reply to Reviewer

8) Thank you for pointing it out. We revised the word from “sold” to “used”. Please confirm it.

- “In Japan, this stretcher has already used to hospitals, nursing homes, ambulance services, schools, police departments, the Self-Defense Forces, and other institutions.”

Reviewer 2 comments

#2-1. Reviewer-2

1. The introduction was well-written and focused on the key aspects of the relevant background information for the study. One point of minor confusion may be lines 56-58, where the authors discuss the potential for falls, and how dropping the patient may lead to severe injuries. I would suggest including a sentence or clarifying here – given that incidence of drops/falls is not part of the present study, how does this relate to the physical burden of traditional devices versus the Airstretcher? Would the Airstretcher likely have a lower drop rate?

Reply to Reviewer

1) We wish to express our appreciation to the Reviewer for their insightful comments, which have helped us significantly improve the paper. As the reviewer gave us the suggestion, the incidence of drops/falls is not part of the present study. However, as we mentioned in the introduction, transporting patients down stairs is associated with a particularly high fall risk for patients and the occurrence of back pain among EMTs. We believe the Airstretcher device has not only the benefit of reducing physical burden but also reducing the risk of fall events.

We added the sentence in the introduction as follow: 

- “This device has the benefit of reducing the risk of fall events by avoiding the need to lift the patient.”

#2-2. Reviewer-2

2. In the Methods Section (lines 77-79), this portion appears to be a bit repetitive. The specific details provided in this paragraph could likely be combined with the above sentences, to reduce unnecessary repetition.

Reply to Reviewer

2) Thank you for pointing out. We organized these sentences. Please confirm the sentences in the Methods section.

#2-3. Reviewer-2

3. In the Methods Section (line 133), it may be helpful to clarify further how the pulse rate was measured. This may help address questions about variability in the pulse rate measurements, etc.

Reply to Reviewer

3) Thank you for helpful suggestion. We revised the sentence as follow: 

- "Additionally, pulse rates after the task in all participants were measured [8] using fingertip pulse oximeter. The Borg CR10 scale, pulse rates and the carrying duration were recorded and put it into the database by researchers.".

#2-4. Reviewer-2

4. The Results Section was very thorough, and provided detailed information about the outcomes of the study. One minor concern throughout was the readability – while minor, it may help with the ‘flow’ of the results section to simplify the naming of the four conditions with acronyms. In lines 152-153 where the four conditions are listed, parentheses could be included after each condition with an abbreviation or acronym. This would help reduce some of the longer sentences/phrases later in the results section, and clarify the overall section for the reader.

Reply to Reviewer

4) Thank you for providing these insights. We expressed the name of four devices as AD (Airstretcher by dragging), AL (Airstretcher by lifting), BL (Backboard by lifting) and TL (Tarpaulin by lifting) in the Result section to reduce some of the longer sentences. However, in other section, we don't express these devices as AD, AL, BL and TL to avoid confusing for readers. Please confirm these sentences in the Result section.

#2-5. Reviewer-2

5. Also in the Results Section, it may help to ‘break up’ the Borg RPE section with a few subheadings. This would clarify to the reader what specific aspect of the Borg RPE reporting that the current section is addressing.

Reply to Reviewer

5) Thank you for your helpful recommendation. We inserted sub-headings in the Borg section as "Overall comparisons", "Sub-group analysis", Interaction between the Borg CR10 scale and other factors", "Head position" and "Toe and side positions". Please confirm these sub-headings in the Borg section.

#2-6. Reviewer-2

6. The Discussion Section was very well done, and easy to follow. One minor point would be the sentence in lines 247-248 dealing with muscle oxygenation. For those familiar with the relationship between muscle oxygenation and physical exertion/workload, this sentence does provide further evidence of the physical demands of transporting patients. However, this sentence may be more clear if the authors expanded upon the interpretation of the findings that muscle oxygenation differs for trained versus untrained individuals, given that the nature of this paper may interest readers who are not as familiar with concepts such as skeletal muscle oxygenation.

Reply to Reviewer

6) Thank you for your kind suggestion. We inserted the sentence of "Therefore, muscle oxygenation may differ between trained and untrained individuals." in the last of Discussion section.

- An experimental study revealed that increased muscle oxygenation for experienced subjects is less than that for novice subjects [10-13]. Therefore, muscle oxygenation may differ between trained and untrained individuals. Further training in the handling of the Airstretcher may enable safer and less stressful patient transport.

#2-7. Reviewer-2

7. The Limitations portion of the Discussion Section was well-written, and sets up future directions in a concise but meaningful way. This section was particularly well-handled.

Reply to Reviewer

7) Thank you for great comments for us. 

#2-8. Reviewer-2

8. One concern in the Figures – Figure 2 presents a box-and-whisker plot for the Borg RPE values. However, this Figure is a bit unclear given the alignment of the dots and variability of their placement. It may help clarify the figure to align the dots over the condition they represent, or choose some other way to represent the individual data.

Reply to Reviewer

8) Thank you for your helpful recommendation. We revised Fig 2, 4 and 5.

Lastly, I wish to mention, Thanks to editor and reviewers.

Yours sincerely,

---

## [Decision Letter · Decision Letter 1]

1 Sep 2022

Use of the Airstretcher with dragging may reduce rescuers’ physical burden when transporting patients down stairs

PONE-D-21-35282R1

Dear Dr. Takei,

We’re pleased to inform you that your manuscript has been judged scientifically suitable for publication and will be formally accepted for publication once it meets all outstanding technical requirements.

Kind regards,

Denis Alves Coelho, PhD

Academic Editor

PLOS ONE

Additional Editor Comments (optional):

Reviewers' comments:

Reviewer's Responses to Questions

**Comments to the Author**

1. If the authors have adequately addressed your comments raised in a previous round of review and you feel that this manuscript is now acceptable for publication, you may indicate that here to bypass the “Comments to the Author” section, enter your conflict of interest statement in the “Confidential to Editor” section, and submit your "Accept" recommendation.

Reviewer #1: All comments have been addressed

Reviewer #2: All comments have been addressed

2. Is the manuscript technically sound, and do the data support the conclusions?

Reviewer #1: Yes

Reviewer #2: Yes

3. Has the statistical analysis been performed appropriately and rigorously? 

Reviewer #1: Yes

Reviewer #2: Yes

4. Have the authors made all data underlying the findings in their manuscript fully available?

Reviewer #1: Yes

Reviewer #2: Yes

5. Is the manuscript presented in an intelligible fashion and written in standard English?

Reviewer #1: Yes

Reviewer #2: Yes

6. Review Comments to the Author

Reviewer #1: (No Response)

Reviewer #2: All my comments were addressed completely and thoroughly, and I have no further comments or revisions to suggest.

7. PLOS authors have the option to publish the peer review history of their article (what does this mean?). If published, this will include your full peer review and any attached files.

Reviewer #1: **Yes: **Arthur de Sá Ferreira

Reviewer #2: No

---

## [Editor Report · Acceptance letter]

5 Sep 2022

PONE-D-21-35282R1 

Use of the Airstretcher with dragging may reduce rescuers’ physical burden when transporting patients down stairs 

Dear Dr. Takei:

I'm pleased to inform you that your manuscript has been deemed suitable for publication in PLOS ONE. Congratulations! Your manuscript is now with our production department. 

Kind regards, 

on behalf of

Dr. Denis Alves Coelho 

Academic Editor

PLOS ONE